# Does Nursing Behaviour of Sows in Loose-Housing Pens Differ from That of Sows in Farrowing Pens with Crates?

**DOI:** 10.3390/ani12020137

**Published:** 2022-01-07

**Authors:** Dierck-Hinrich Wiechers, Swetlana Herbrandt, Nicole Kemper, Michaela Fels

**Affiliations:** 1Institute for Animal Hygiene, Animal Welfare and Farm Animal Behaviour, University of Veterinary Medecine Hannover, Foundation, Bischofsholer Damm 15, 30173 Hannover, Germany; Nicole.Kemper@tiho-hannover.de (N.K.); Michaela.Fels@tiho-hannover.de (M.F.); 2Statistical Consulting and Analysis Center for Higher Education, TU Dortmund University, Vogelpothsweg 87, 44221 Dortmund, Germany; Swetlana.Herbrandt@tu-dortmund.de

**Keywords:** suckling, nursing, loose-housing, sow behaviour, maternal behaviour, free-farrowing

## Abstract

**Simple Summary:**

Around birth and during lactation, sows are often confined to narrow farrowing crates in order to restrict their movements to prevent piglet crushing by sows. Since these housing systems are increasingly questioned for sow welfare reasons, more research is being carried out on alternatives such as loose-housing farrowing systems. As one central aspect of sows’ maternal behaviour, nursing behaviour impacts the mortality rate and growth performance of the piglets. Since maternal behaviour is, in turn, affected by the housing environment, this study aimed to compare a farrowing pen without sow confinement to one with a crate regarding the nursing behaviour of sows. The study results underline that sows’ nursing behaviour was influenced by the housing system and, furthermore, that the nursing behaviour of loose-housed sows was more similar to that of sows in semi-natural conditions. Unconfined sows terminated more nursing bouts and nursed piglets for a shorter period. The nursing frequency decreased in the course of lactation in both systems, whereas the proportion of sow-terminated nursing bouts increased in the four-week examination period. These might be the first steps in the natural weaning of piglets.

**Abstract:**

Sows confined to farrowing crates are restricted in performing natural behaviour such as maternal behaviour. Loose-housing farrowing pens (LH) and farrowing pens with crates (FC) were compared regarding sows’ nursing behaviour via video analyses over four weeks per batch (one day per week). Nursing frequency was similar in LH and FC pens (1.25 ± 0.82 vs. 1.19 ± 0.75 nursings/sow/hour; *p* > 0.05). However, nursing duration differed between the two systems (LH: 5.7 ± 4.6 min vs. FC: 7.0 ± 5.0 min; odds ratio (OR) 1.168, *p* = 0.011). In LH pens, more nursing bouts were sow-terminated than in FC pens (OR 0.427, *p* = 0.001). The probability of sow-terminated nursing occurring increased from week 1 to week 4 (OR 3.479, adjusted *p* (*p*_adj_) < 0.001), while that of observing unnursed piglets decreased from week 1 to week 4 (OR 0.301, *p*_adj_ < 0.001) and rose with increasing litter size (OR 1.174, *p* = 0.010). We conclude that nursing behaviour was affected by the farrowing system, with shorter nursing duration and more nursing terminations by the sow in LH than in FC pens. Since this corresponds to the nursing behaviour of sows in semi-natural conditions, it can be assumed that sows in LH pens are more likely to exhibit natural nursing behaviour.

## 1. Introduction

Farrowing crates for sows were widely introduced in the 1960s to reduce crushing of piglets by sows [1]. Although it is increasingly recognised that farrowing crates are not compatible with sow welfare [2], crate systems are still common in conventional pig farms [3,4]. While the use of farrowing crates is already prohibited in some European countries such as Switzerland, Sweden and Norway [5], the EU has recently announced to draft a legislative proposal, aimed to phase out and finally forbid the use of farrowing crates as well [6].

To improve sow welfare, alternatively, crateless farrowing systems have been developed in recent years, and more research has been carried out on the effects of these systems on sows and piglets. While there are already several studies on the occurrence of piglet crushing as well as on the behaviour of sows and piglets in farrowing systems without crates [7,8,9,10,11], scientific knowledge concerning the effects of such alternative farrowing systems on the nursing behaviour of sows and suckling behaviour of piglets is still lacking.

Crushing and starvation or a combination of both, with crushed malnourished and thus weakened piglets, are the most frequent causes of piglet loss related to the farrowing area [3,12,13], subsequently resulting in a significant economic loss in the pig industry. Furthermore, mortality due to crushing or malnutrition is an important piglet welfare issue [3,14]. Among other reasons, such as the low birth weight of piglets, the sow’s nursing behaviour can play a role in this regard, as starvation is directly linked to piglets’ insufficient milk intake, whereas piglet crushing may be indirectly linked to it due to subsequent piglet weakness. Apart from that, well-functioning maternal behaviour, which is related to the sows’ nursing behaviour, is important in order to obtain fast and evenly growing litters during the lactation period [15,16].

A domestic sow has 20 or more nursing bouts per day under commercial conditions. Nursing is accompanied by specific grunt vocalisations by the sow and can be divided into five phases: 1. The piglets arrange themselves at the udder and choose their teat (this lasts for a few seconds to several minutes); 2. piglets massage the udder (this usually takes about one minute); 3. piglets suckle slowly (for about 20 s); 4. fast suckling with milk ejection (this phase only lasts 10–20 s with an intake of about 50 g milk per piglet); 5. finally, slow suckling or nosing of the udder. The last phase is as variable in time as the first one, depending on whether piglets leave the udder directly, continue nosing or fall asleep [17,18]. Udder massaging by the piglets activates a neuro-hormonal reflex. Oxytocin-secreting neurons of the hypothalamus are activated, and oxytocin is released into the blood. Myoepithelial cells surrounding the alveoli contract, with an increase in intramammary pressure resulting in milk let-down [19,20].

In order to reduce piglet losses and obtain favourable piglet growth performance in farrowing units, more attention should be paid to the maternal characteristics of sows. Maternal behaviour seems to be less pronounced in gilts and has yet to develop over at least two parities [7,21]. Additionally, there seems to be a large variation in maternal or individual behaviour of sows [22]. Good maternal behaviour in sows can be achieved through breeding and is considered essential for the successful establishment of free-farrowing pens [2,14]. However, the housing system also has an impact on sows’ maternal behaviour. For instance, in get-away pens, a stronger or faster reaction of sows towards screaming piglets was observed compared to single- or crate-housed sows, which resulted in both lower piglet crushing losses and total piglet losses in get-away pens [7,23]. Nonetheless, there is still a lack of knowledge concerning the effects of different farrowing systems on the sows’ nursing behaviour.

The aim of this study was to investigate the nursing behaviour of sows in two different farrowing systems as one central aspect of maternal behaviour, which affects both piglet mortality and growth performance [15,16]. It was hypothesised that nursing behaviour in loose-housing pens is not impaired compared to farrowing pens with crates, precisely because an intensive sow-piglet contact is possible. Therefore, the nursing bouts were studied in depth for sows in farrowing pens with crates and for sows in a loose-housing system without crates. Nursing frequencies, durations of individual nursing bouts, the number of sow-terminated nursing bouts and the number of unnursed piglets per nursing bout were analysed by video observation during the housing period.

## 2. Materials and Methods

### 2.1. Animals, Housing and Handling

The study was carried out between June and December 2018 on the research farm of the Chamber of Agriculture of Lower Saxony in Bad Zwischenahn-Wehnen, Germany. The experiments were conducted in accordance with both European (Directive 2008/120/EC) and German legislation (Tierschutzgesetz (Animal Protection Act) and Tierschutz-Nutztierhaltungsverordnung (Animal Protection—Livestock Ordinance)). The Animal Welfare Officer of the University of Veterinary Medicine Hannover, Foundation, Germany reviewed and approved the study beforehand. No invasive procedures were performed on the animals as part of these investigations.

Two different farrowing systems were investigated: a loose-housing system without farrowing crates (LH), with six housing units per room, and a conventional system with farrowing crates (FC), with eight housing units per room. LH and FC pens were installed in neighbouring rooms of the same building and were subject to the same management procedures. Both systems were developed and produced by the same manufacturer (Big Dutchman International AG, Vechta, Germany). Per batch, six sows were studied in each farrowing system. A total of five consecutive batches were included in the study with a total of 60 sows (Landrace × Large White, d.b. Victoria, BHZP GmbH, Dahlenburg-Ellringen, Germany), from first to seventh parity (LH: 3.8 ± 1.6, FC: 4.1 ± 1.7) and their offspring.

The FC pen’s dimensions were 260 cm in length and 200 cm in width (5.2 m^2^). The 190 cm-long and 80 cm-wide farrowing crate was located in the centre of the pen and provided a usable area of 1.52 m^2^ to the sow. The 150 cm-long and 60 cm-wide, three-sided open creep area for piglets was arranged parallel to the sow’s crate. The creep area was heated by a 150 W infrared lamp as well as by a heated concrete floor (Figure 1).

The 250 cm-long and 240 cm-wide (6 m^2^) LH pen had 4.01 m^2^ usable space for the sow. The creep area for piglets was separated from the sow’s area by a swivelling iron grid with which the sow could be confined if necessary. The 125 cm × 75 cm-large creep area was open on two sides and was equipped with a 150 W infrared light as in the FC system. The floor area was covered with a solid rubber mat. An anti-crushing bar was installed as a mushroom-shaped protrusion at the long side of the pen, while piglet protection bars were located at the shorter, open sides. Both were installed to protect the piglets from being crushed by the sow (Figure 2).

The two farrowing systems had the same slatted plastic flooring (10 mm gaps and 11 mm slats) with a tiled and non-perforated lying area for the sow. In both systems, sows were offered a jute sack from the time of entering the pen to enable nest-building behaviour on the days before farrowing. As further manipulable material, a large cotton rope was hung up for the sow and a small one for the piglets. A rack with hay was additionally installed in the LH pen. If necessary, consumed manipulable material was replenished.

Before entering the farrowing pens, the pregnant sows were kept in groups of three to five in another compartment in the same building. Five days before the expected farrowing, the sows were taken to the farrowing systems where they were single-housed with (FC) or without crates (LH). The sows were randomly assigned to the farrowing systems when the study began. Afterwards, they were always kept in the same system. While FC sows were confined to the crate throughout the study period, LH sows were not at any time. All sows received a commercial lactation diet twice per day (07:30 and 16:30). Feed quantity was rationed: a maximum of 5 kg on the days before farrowing and a maximum of 2 kg on the day of birth. Afterwards, it was increased by about 0.5 kg per day to reach ad libitum feeding after about 14 days (8–9 kg). After a 33-day period in the respective farrowing systems, sows left the pen, were brought to the service centre and a new reproductive cycle began. Within 24 h after the piglets’ birth, they were ear-tagged for identification and their canines were shortened to avoid injuries to the mother’s udder. Male piglets were castrated on the third day of life. On the same day, the piglets’ tails were docked. Within the first three to 72 h after the piglets’ birth, cross-fostering was performed within the same farrowing system in order to achieve a uniform litter size of about 14 piglets. From the tenth day of life until the end of the housing period in the farrowing systems, the piglets in both systems were fed a commercial weaning supplementary feed ad libitum. Therefore, small plastic feeding bowls were used. The LH-system as well as the FC-system were illuminated from 07:30 to 17:30. At night, a dimmed light regime was used. When they were 28 days old, the piglets were weaned and transferred to the farm’s own rearing unit.

### 2.2. Video Analysis

A video system was set up to record the animals’ behaviour. Centrally located above the pens, cameras (Everfocus ez.HD, Everfocus, New Taipei City, Taiwan) were installed to observe the entire pen area. These were connected to a digital video recorder (Everfocus ECOR FHD 16 × 1, Everfocus, New Taipei City, Taiwan) and videos were continuously recorded on hard drives throughout the experimental period.

The behaviour of the sows and their piglets was analysed in five batches, one day per week by the same observer in order to evaluate it over the course of the lactation period. Saturdays were selected for the eight-hour observation periods, since on weekends, there was little disturbance for the animals in the barn to enable standardised examination conditions. Thus, four days per sow were analysed in each batch. The day was divided into a morning (06.00–10:00) and an afternoon (13:00–17:00) period, lasting four hours each. Due to technical problems, there was a small loss of video material, resulting in missing data of ten sows from both systems (LH: *n* = 4; FC: *n* = 6) for some hours in the early morning of one observation day in batch 2, and of one sow in the afternoon of one observation day in the same batch.

In the above-mentioned time slots, the sows’ nursing behaviour was analysed in detail. The total number of nursing bouts was determined for each sow during each observation period. The duration of each nursing bout (in seconds) was recorded by determining the start time and end time of the nursing bout. In accordance with previous studies (e.g., [7,8,24]), the start of a nursing bout was defined as the time when at least 75% of the piglets present in a litter gathered around the sow’s udder, found their teat and started massaging the udder. This required the sow to lie laterally on her side, stretching out all four limbs and offering her teats to the piglets. The end of a nursing bout may be determined by either the sow or her piglets. A sow-terminated nursing bout ended when she changed her body position so that the udder was no longer accessible to the piglets. This was the case either when rolling into a prone position or when sitting up or standing. The end of a regular, piglet-terminated suckling bout was defined by at least 75% of the piglets in the litter having left the udder or fallen asleep on it. In order to define these occasions, the current number of piglets present in a litter (litter size) was determined for each day and each sow individually. Thus, it was also possible to establish whether all piglets were at the udder to suckle or if piglets remained unnursed during a nursing bout. The latter was the case if a piglet had not suckled once by the end of a nursing bout.

### 2.3. Statistical Analysis

Statistical analyses were performed using the statistics software R [25]. Levels of significance were set at *p* < 0.05.

#### 2.3.1. Nursing Frequency

A Poisson model was used to analyse the number of nursing bouts per sow per hour (nursing frequency) by implementing the R package ImerTest [26]. Farrowing system, sows’ parity, observation day, time of day (morning/afternoon) and the litter size were set as fixed effects. Sow, pen and batch were considered as random effects. Pairwise comparisons of all observation days were carried out using the Wald test by applying the R package emmeans [27]. Resulting *p*-values were adjusted using the Bonferroni-Holm method [28]. As coefficient of determination, pseudo *R*^2^ was calculated for the model by using the delta-method and the R package MuMIn [29].

#### 2.3.2. Sow-Terminated Nursing and Unnursed Piglets

A logistic regression model with a logit-link function was used to model both the probability of observing sow-terminated nursing bouts in the two farrowing systems (yes/no) and the probability of observing piglets that remained unnursed (yes/no). Therefore, the R package ImerTest [26] was used. In this model, farrowing system, sows’ parity, observation day, time of day (morning/afternoon) and the litter size were set as fixed effects, while sow, pen and batch were set as random effects. In order to compare the different observation days, pairwise comparisons were made using the Wald test and the R package emmeans [27]. The Bonferroni-Holm method was used to adjust the *p*-values [28]. By using the delta-method and the R package MuMIn, pseudo *R*^2^ was determined for both models [29].

#### 2.3.3. Duration of Nursing Bouts

For analysing nursing durations, a gamma regression model with log link was used. Farrowing system, sows’ parity, observation day, time of day (morning/afternoon) and the litter size were considered as fixed effects. For all these effects, the interaction with sow-terminated suckling (yes/no) was also analysed. Sow, pen and batch were set as random effects. Based on the model, the durations of nursing bouts were compared by pairwise post-hoc analyses for the two farrowing systems, depending on the presence or absence of a sow termination. For this purpose, a Wald test was performed using the R package emmeans [27]. To detect any differences in the nursing durations between the times of day (morning and afternoon), the same procedure was applied. To reveal any differences between the four observation days, multiple pairwise comparisons of all observation days were conducted. All days were compared depending on the presence or absence of a sow termination using the Wald test. The R package emmeans [27] was used for this purpose. All resulting *p*-values were adjusted using the Bonferroni-Holm method [28] and pseudo *R*^2^ was determined by using the delta-method and the R package MuMIn [29].

## 3. Results

### 3.1. Nursing Frequencies

In total, 1363 nursing bouts were analysed for LH sows and piglets, and 1304 nursing bouts were analysed for sows and piglets in FC pens. Concerning the nursing frequency, there was no significant difference between the LH system and the FC system (1.25 ± 0.82 vs. 1.19 ± 0.75 nursing bouts per sow per hour, OR 0.963, confidence interval (CI) [0.882, 1.051], *p* = 0.397) (Table 1). While the litter size (OR 0.998, CI [0.978, 1.018], *p* = 0.831), sows’ parity (OR 1.000, CI [0.975 1.025], *p* = 0.999) or the time of day (OR 1.003, CI [0.930, 1.082], *p* = 0.939) had no influence on the nursing frequency, there was an effect of the observation day. From observation day 1 in the first week after farrowing (1.27 ± 0.80 nursings/sow/hour) to observation day 4 in the fourth week after farrowing (1.06 ± 0.70 nursings/sow/hour), nursing activity per hour decreased significantly (OR 0.835, CI [0.746, 0.934], *p*_adj_ = 0.008). It also decreased from observation day 2 (1.37 ± 0.86 nursings/sow/hour) to observation day 3 (1.18 ± 0.74 nursings/sow/hour, OR 0.859, CI [0.774, 0.954], *p*_adj_ = 0.018). The decrease between day 2 and day 4 was highly significant (OR 0.771, CI [0.691, 0.860], *p*_adj_ < 0.001).

For the model, a pseudo R^2^ of 0.011 was calculated.

### 3.2. Nursing Termination by the Sow

In the LH pens, 65.3% of the nursing bouts were terminated by the sow, whereas in the FC pens, 58.2% were sow-terminated (OR 0.427, CI [0.255, 0.716], *p* = 0.001) (Figure 3).

While parity did not affect the occurrence of sow-terminated nursing bouts (OR 1.089, CI [0.924, 1.283], *p* = 0.310), the litter size (OR 1.109, CI [1.008, 1.220], *p* = 0.033) and the time of day (OR 1.300, CI [1.095, 1.543], *p* = 0.003) were shown to have an effect. In the morning, 59.5% of the nursing bouts were sow-terminated, whereas this figure rose to 64.03% in the afternoon. In Figure 4, the results of the statistical model for the probability of sow-terminated nursing are shown for different litter sizes.

The probability of sow-terminated nursing occurring rose with an increasing litter size. The observation day affected the occurrence of sow-terminated nursing bouts as well. Compared to observation day 1, the probability of nursing termination by the sow occurring increased for every consecutive observation day (all OR > 2.139, all CI [1.682, 4.569], all *p*_adj_ < 0.001). From day 2 (OR 1.626, CI [1.262, 2.096], *p*_adj_ < 0.001) and day 3 (OR 1.412, CI [1.089, 1.830], *p*_adj_ = 0.019) to day 4, there was also a significant increase (Table 2).

A pseudo *R*^2^ of 0.220 was determined for this model.

### 3.3. Piglets Left Unnursed

In 8.5% of the nursing bouts of LH sows and 6.6% of those of FC sows, at least one piglet remained unnursed (Table 3). The odds of having unnursed piglets in a nursing bout were not affected by the farrowing system (OR 0.626, CI [0.376, 1.043], *p* = 0.072), parity (OR 0.975, CI [0.832, 1.142], *p* = 0.750) or time of the day (OR 0.738, CI [0.541, 1.007], *p* = 0.055).

However, it was shown that when litter size increased, the probability of unnursed piglets occurring increased as well (OR 1.174, CI [1.039, 1.326], *p* = 0.010) (Figure 5).

From day 1 (OR 0.301, CI [0.172, 0.525], *p*_adj_ < 0.001) and day 2 (OR 0.386, CI [0.223, 0.669], *p*_adj_ = 0.003) to day 4, the probability of observing unnursed piglets decreased, whereas just a tendency to decrease was apparent from day 3 to 4 (OR 0.485, CI [0.273, 0.862], *p*_adj_ = 0.055).

The pseudo *R*^2^ for this model was 0.057.

### 3.4. Duration of Nursing Bouts

For LH sows, a nursing bout took 5.69 ± 4.56 min, for FC sows, 7.01 ± 4.96 min (OR 1.168, CI [1.036, 1.317], *p* = 0.011) on average. Sow-terminated nursing bouts took 3.49 ± 2.56 min for LH sows and 3.95 ± 2.59 min for FC sows (OR 1.206, CI [1.082, 1.344], *p*_adj_ = 0.001). In cases of piglet-terminated suckling, a bout lasted 9.83 ± 4.62 min in the LH pens and 11.27 ± 4.27 min in the FC pens (OR 1.168, CI [1.036, 1.317], *p*_adj_ = 0.011) (Table 4).

A sow-terminated nursing bout was associated with a significant reduction in nursing duration in both the LH system and the FC system (all OR < 0.359, all CI [0.325, 0.384], all *p*_adj_ < 0.001) compared to a piglet-terminated suckling bout. This effect was found for all observation days (all OR < 0.460, all CI [0.265, 0.500], all *p*_adj_ < 0.001). Nursing bouts terminated by piglets showed fairly constant durations on the different days of observation, except for a significant difference (OR 1.210, CI [1.092, 1.341], *p*_adj_ = 0.002) between day 1 (9.88 ± 4.21 min) and day 4 (12.08 ± 4.90 min) (Table 2). In cases of sow-terminated nursing bouts, the nursing duration was longer at day 1 than at day 2 (OR 0.771, CI [0.714, 0.833]), day 3 (OR 0.828, CI [0.766, 0.896]) and day 4 (OR 0.769, CI [0.709, 0.833]) (all *p*_adj_ < 0.001). Neither with (OR 1.040, CI [0.987, 1.095]) nor without sow termination (OR 0.963, CI [0.900, 1.029], all *p*_adj_ = 0.289) was a significant difference found in the nursing duration between morning (6.50 ± 4.89 min) and afternoon (6.18 ± 4.72 min). However, the presence of sow-termination significantly reduced the expected duration of nursing both in the morning (OR 0.340, CI [0.320, 0.362]) and in the afternoon (OR 0.367, CI [0.345, 0.391], all *p*_adj_ < 0.001) compared with piglet-terminated suckling bouts.

While the number of suckling piglets per litter had no influence on nursing duration per se (OR 1.005, CI [0.978, 1.032], *p* = 0.738), there was a significant effect of the interaction between the litter size and the occurrence of sow-termination (OR 0.974, CI [0.951, 0.998], *p* = 0.031). The shorter the duration of sow-terminated nursing bouts the more piglets were present in a litter. In contrast, the duration of piglet-terminated nursing bouts rose when the litter size increased (Figure 6).

The sow’s parity had no influence on nursing duration (OR 1.011, CI [0.974, 1.049], *p* = 0.559), with no observed difference between sow- and piglet-terminated nursing duration (OR 0.972, CI [0.944, 1.001], *p* = 0.058).

The pseudo *R*^2^ for this model is calculated with 0.469.

## 4. Discussion

### 4.1. Nursing Frequencies

Under semi-natural conditions, a nursing frequency of 1.3 nursing bouts per sow per hour was determined for the first days after birth [30], which is nearly the same frequency that was found on the first observation day in the present study (1.27 ± 0.80 nursings/sow/hour, i.e., one nursing bout per 47.4 min). In general, one nursing bout every 40–60 min is considered a normal frequency [20], which is in line with our observations in the current study. The nursing frequency plays a central role for the milk output of the sow and thus milk intake of the piglets. Although it is known that the number of nursing bouts without milk flow goes up with increasing nursing frequency, overall, more milk is ejected when the nursing frequency rises and, consequently, litters gain more weight [31]. It was therefore an interesting question whether the nursing frequency differs depending on whether the sow is restrained in a crate or not.

However, there was no difference in nursing frequency between the two farrowing systems investigated in the present study. Thus, housing sows in single loose-housing pens does not seem to impair nursing activity in general. It was even reported in an earlier study that loose-housed sows spent a greater proportion of time nursing than crated sows [32], but our study did not provide any indications for this. Even if there have only been few studies on this topic so far, our results agree with some earlier studies that also found no differences in nursing frequency between loose-housed and crated sows [7,24]. Thodberg et al. [7] only analysed two days in the early stage of lactation, whereas in the present study, the entire lactation period of four weeks was investigated. In the study by Singh et al. [24], sows farrowed in a conventional farrowing pen with a crate and did not enter the loose-housing pen until day 3 of lactation. The nursing behaviour was only analysed until day 18 of the lactation period.

Furthermore, the results of the present study did not reveal any effects of the sows’ parity, time of day or litter size on the nursing frequency. However, it was shown that there was an effect based on the observation day, there being a significant decrease in nursing frequency during the four-week period of lactation.

According to Jensen [30], the behaviour of domestic pigs kept under semi-natural conditions is still quite similar to that of wild boars. In a semi-natural environment, domestic sows separate from the group prior to farrowing and build a nest to farrow in. The sow and the litter then remain in or near the nest for about nine days before returning together to the group [30]. This natural behaviour is also reflected in sows kept in farrowing housing systems on farms. While the time around birth is characterised by increased lateral lying [10,21], sows become more active again in the course of the lactation period [16]. Analogous to this behaviour, the nursing frequency decreases significantly when the four-week lactation period progresses, as reported by Jensen [33] for free-ranging domestic sows. A decrease in nursing frequency seems to be more pronounced in sows that can get away from their piglets, with a 30% lower frequency at day 27 after birth compared to sows housed in pens without the possibility of retreating from their piglets [34]. Nevertheless, in the study by Weary et al. [34], nursing frequency declined slightly during the course of lactation for sows in conventional pens as well. This was confirmed by Moreira et al. [35], who found a decrease in nursing frequency in early lactation from day 7 to 15 after birth in crated sows. In the aforementioned study, the nursing interval increased from 30 to 34.9 min, whereas in the current study, the interval initially decreased from 47.4 to 43.7 min in the second week. Thereafter, the nursing interval rose in the present study as well (50.8 min in the third week and 56.7 min in the fourth week). Other studies also revealed an initially increasing nursing frequency in the early lactation period [36,37], as the sow’s milk yield rises rapidly at this time [37].

While at the beginning of the lactation period, milk and colostrum intake are essential for piglet vitality [38], a decreasing nursing frequency thereafter, with reduced milk availability, motivates the piglets to forage on their own [5]. The decreasing nursing activity aimed to reduce milk availability for the piglets can already be considered as part of the weaning process, which is gradual and continuous under semi-natural conditions. Piglets are finally weaned by, on average, 17.2 weeks after farrowing [39]. However, it is assumed that some aspects of the weaning process already start in the first days after birth, as some behavioural changes that finally may lead to the piglets being weaned obviously begin then. While the nursing frequency tended to increase within 10 days post farrowing, which is in line with our results, the semi-naturally kept sows initiated a decreasing proportion of the nursing bouts and terminated more of them [36].

By reducing milk availability through these mechanisms, the sow saves body reserves for the following reproductive cycle [5,7]. However, it has to be emphasised that during the four-week lactation period in the present study, the average never dropped below one nursing bout per sow per hour. Thus, the sows just reduced the nursing frequency slightly, ensuring that the piglets were provided with sufficient milk.

### 4.2. Nursing Termination by the Sow

While during the first week of lactation, there was still a similar ratio of sow-terminated and piglet-terminated nursing bouts in the LH pens, more nursing bouts in the FC pens were piglet-terminated than in LH pens. Meanwhile, more nursing terminations by the sow were observed in both systems from week 2 of lactation (Figure 3).

Even in a semi-natural environment, sow-terminated nursing bouts gradually increased to about 60% until day 10 post farrowing [36], which is very similar to the results of the current study. Jensen and Recén [39] observed in free-ranging domestic sows, an increase up to almost 100% in week 4 of lactation, whereas in week 1, only about 40% of nursing bouts were sow-terminated. An increase in sow-terminated nursing bouts was also observed in conventional systems (FC and LH systems), taking into account only the period from day 4 to day 18 of lactation [24]. Valros et al. [37] also found a significant increase in sow-terminated nursing bouts during a 30-day study period to around 65% of all nursing bouts. However, only sows in LH pens were investigated. As mentioned above, sows reduce the amount of milk they give to their piglets in the course of lactation. This is not only regulated by the frequency of nursing, but also by the termination of ongoing nursing bouts to prevent the final massage of the udder. The latter stimulates blood-flow through the udder and promotes subsequent milk production. Furthermore, prolactin release is initiated in this way [5,40], which causes the maintenance of milk production during the lactation period [41].

A significant increase in nursing terminations by the sow in the course of lactation was also shown in the present study. Furthermore, it was revealed that sows in the LH system (65.3%) terminated nursing activity significantly more often than sows in the FC system (58.2%). This agrees with some previous studies investigating the nursing behaviour of gilts and sows. According to Thodberg et al. [7], a higher degree of control over nursing behaviour may be reflected in increased nursing termination frequencies of gilts in loose-housing pens in contrast to crated gilts. Similar findings were reported for loose-housed sows compared to crated sows some days after farrowing [42]. Crated sows are generally more restricted in the expression of their natural behaviour than sows in pens without crates [14,43]. As mentioned above, under semi-natural conditions, sows also leave the nest and get away from their piglets [30], which can be interpreted as piglet-avoidance behaviour of sows [44]. Crated sows are confined in close proximity to their piglets. The ability of behavioural control over this situation is reduced, whereas sows in loose-housing pens are more able to reduce their maternal investment over lactation. The impossibility of escaping from the piglets may cause stress in crated sows [5]. In fact, some studies have shown increased stress levels in confined sows [45,46,47]. However, sows housed in the two farrowing systems in the present study had comparable cortisol levels measured in their hair [48].

Apparently, an enriched environment (such as the offer of straw in farrowing pens without crates) already seems to be sufficient to induce an increased frequency of nursing termination by the sow compared to a more barren environment (loose-housing pens without straw) [49]. In the LH system in the current study, sows were also offered hay. This may have influenced nursing behaviour as well, perhaps resulting in more sow-terminated nursing bouts.

Terminating nursing bouts more often is a quite natural behaviour of sows in terms of decreased maternal investment over lactation [5] and may be stimulated by a more natural environment or one that promotes more sow welfare. Furthermore, it could also be assumed that sows in loose-housing pens are generally more active than sows in narrow crates, which could also be reflected in more sow-terminated nursing bouts. However, activity in loose-housing pens does not seem to be increased, since Nicolaisen et al. [10] showed similar body posture change frequencies in loose-housing pens and in farrowing pens with crates. The study by Nicolaisen et al. [10] was carried out on the same farm as the present study, even though the loose-housing pens were modified for the present study.

The results of the current study indicated that the probability of a sow-terminated nursing bout was also affected by the litter size. Interestingly, Illmann et al. [50] found similar results, with a higher probability of nursing terminations by sows with larger litters four days post farrowing. The same study found that teat fights increased with increasing litter size 25 days post-farrowing. The fact that teat fights are influenced by litter size has already been reported by Miligan et al. [51]. The increasing probability of nursing terminations by the sow in the presence of larger litters may be related to the occurrence of teat fights among the piglets, since teat fights promote the termination of nursing by the sow [8]. Fights among piglets and piglets screaming at the udder can irritate sows and lead to unsuccessful nursing when pigs are kept in a semi-natural environment [52]. Thus, piglet behaviour could cause the sows to stop nursing under farming conditions, too.

Furthermore, it was shown in the present study that the time of the day had an effect on the occurrence of sow-terminated nursing bouts, with a higher probability of sow-terminated nursing in the afternoon. To the best of our knowledge, there is no other study that has already examined nursing terminations by the sow depending on the time of day. It was already reported that sows performed more piglet-avoiding behaviour during light hours. The authors conclude that this reflects the increased activity of the piglets in the afternoon, which the sows try to avoid [44]. This could be a reason for more sow-terminated nursing bouts in the afternoon, as found in the present study.

### 4.3. Duration of Nursing Bouts and Piglets Left Unnursed

In the present study, it was found that overall, nursing duration was longer in FC pens than in LH pens, as was also reported by Yun et al. [53], who observed 7.6 min (FC) and 5.7 min (LH) at days 3 and 6 post farrowing, respectively. This is quite similar to our results showing 8.23 min (FC) and 6.25 min (LH) for the first week, although Yun et al. [53] defined a nursing bout starting with 50% of the piglets per litter present at the teats, whereas in the present study, 75% of piglets were required. Moreover, for sow-terminated nursing bouts, nursing duration was longer in FC pens than in LH pens in the current study. This may be due to a reduced ability of crated sows to have control over nursing. According to Thodberg et al. [7], a sow’s higher degree of control over nursing is related to a decreased accessibility to the udder by the piglets, shorter nursing durations, and an increased percentage of sow-terminated nursing bouts at day 10 of lactation. Perhaps sows in crates stopped nursing bouts later due to the limited freedom of movement.

As expected, if nursing was terminated by the sow, the duration of a nursing bout was significantly shortened compared to piglet-terminated suckling bouts in both LH and FC pens. Interestingly, the current study revealed that in the case of piglet-terminated nursing bouts, the duration of nursing bouts was also longer in FC pens than in LH pens, which is in accordance with results by Pedersen et al. [8]. This may be due to the fact that sows confined to crates had lower circulating oxytocin levels, and thus, the piglets had to spend more time stimulating the release of an adequate amount of milk through a longer post-massage [53], since oxytocin triggers contraction of the myoepithelial cells and thus milk ejection [20].

The gradual weaning process might also be reflected in the decreasing duration of sow-terminated nursing bouts from week 1 to week 2, as shown in this study. From week 2 onwards, the duration of nursing bouts remained constant. It is possible that at that time, the sow could not reduce the nursing duration any further and at the same time ensure an adequate milk supply for the piglets. Moreover, a further reduction in post massage of the udder might be impossible. Valros et al. [37] also found a decrease in duration of sow-terminated nursing bouts of loose-housed sows up until day 20 of lactation. Afterwards, the durations of nursing bouts remained constant. Nevertheless, sows in that study nursed considerably longer, with about 7 and 6 min on the two observation days in the first week compared to the sows in the current study, with 4.35 min in the first week post farrowing. However, Valros et al. [37] defined a nursing bout when 50% of the piglets were active at the udder and only counted suckling activity lasting one minute or more.

As highlighted in the present study, there was a significant effect of the interaction between the litter size and the presence of a sow-terminated nursing bout. We found a shorter duration of sow-terminated nursing bouts with increasing numbers of piglets, and an increasing duration of piglet-terminated nursing bouts when the litter size went up. Valros et al. [37] also found that the nursing duration rose with larger litters and concluded that this might be due to an increase in udder stimulation. The opposite seems to be the case for sow-terminated nursing bouts. Larger litter sizes cause more teat fights [51], these leading to sow-terminated nursing [8]. Thus, on the one hand, sows with larger litters may nurse for longer periods when suckling is terminated by the piglets and, on the other hand, sows may terminate nursing bouts faster and more frequently than sows with smaller litter sizes. Which of the two effects is predominant probably depends on the individual characteristics of the sow.

In the current study, it was also shown that in larger litters, the chance of observing nursing bouts with unnursed piglets increased. This may also be related to a high amount of nursing terminations by sows with large litters and to a high amount of teat fights [8]. It can be assumed that competition for the udder also plays a role regarding the amount of unnursed piglets. This is also underlined by the fact that piglets in larger litters have a lower teat consistency [51]. In the present study, 10.9% (first week post-farrowing), 8.5% (second week), 6.9% (third week) and 3.4% (fourth week) of the nursing bouts failed to nurse every piglet in the litter, and at least one piglet remained unnursed. The decrease in the number of nursing bouts with unnursed piglets in the course of lactation could be explained by the fact that the piglets became more active and more vital in the course of lactation on the one hand, and, on the other hand, spent less time in the nest. Previous studies already showed that piglet activity increased with age [30,54]. Consequently, with increasing age, piglets were more aware of the sow’s willingness to nurse and, additionally, they had the vitality to reach the sow’s udder more quickly.

## 5. Conclusions

The results of the current study revealed that the housing system has an impact on sows’ nursing behaviour in general. While LH sows and FC sows had a comparable nursing frequency and a similar percentage of piglets remained unnursed, LH sows terminated more nursing bouts and nursed for a shorter period than FC sows. Consequently, it may be concluded that housing sows in single loose-housing pens impairs nursing activity. However, as these behavioural patterns (i.e., shorter nursing duration and more nursing terminations) seem to be similar to the nursing behaviour of sows in semi-natural conditions [30,33,36,39], it can be assumed that sows in LH pens are more likely to exhibit natural nursing behaviour. This can be interpreted as positive for sow welfare, whereas the effects on the piglets should be the subject of further research.

## Figures and Tables

**Figure 1 animals-12-00137-f001:**
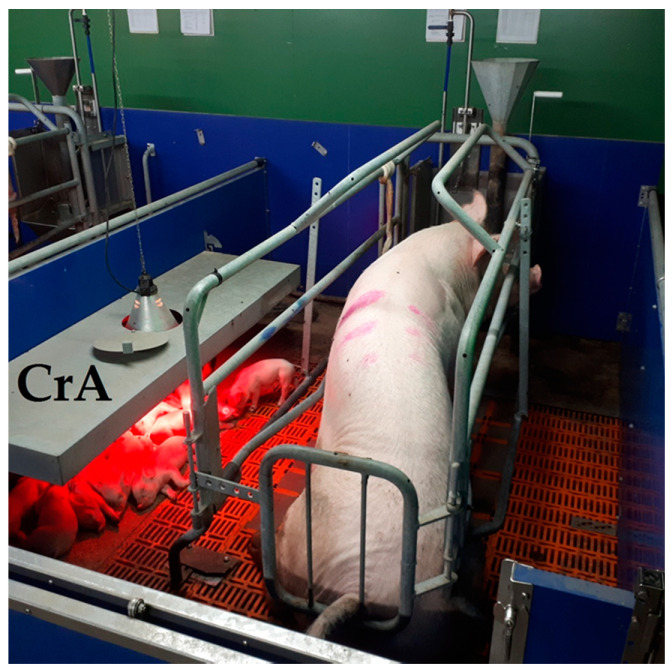
Pen with farrowing crate (FC). CrA, creep area.

**Figure 2 animals-12-00137-f002:**
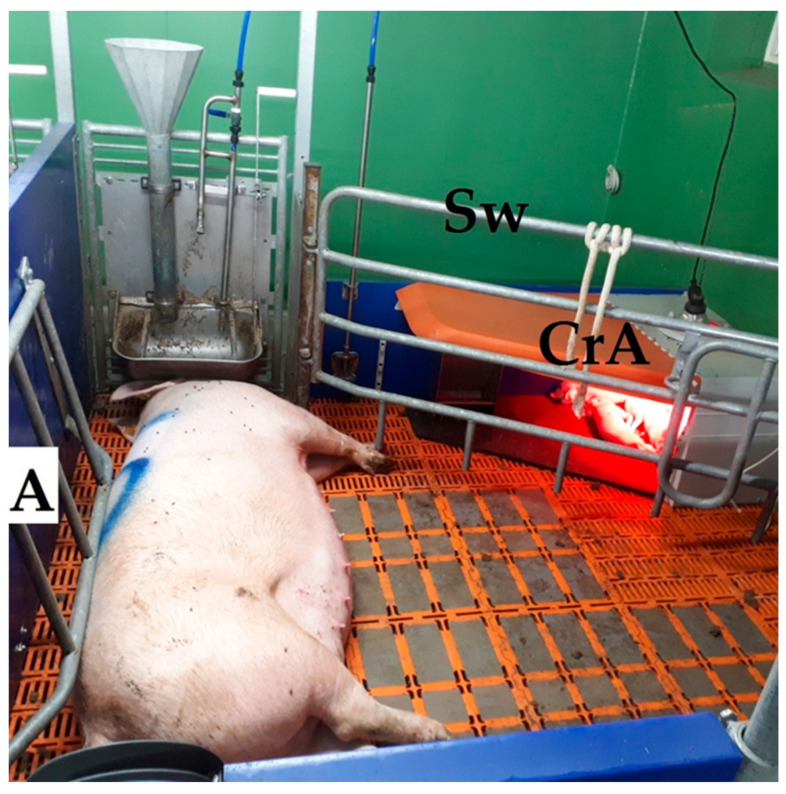
Single loose-housing pen (LH). CrA, creep area; Sw, swing gate; A, anti-crushing bars.

**Figure 3 animals-12-00137-f003:**
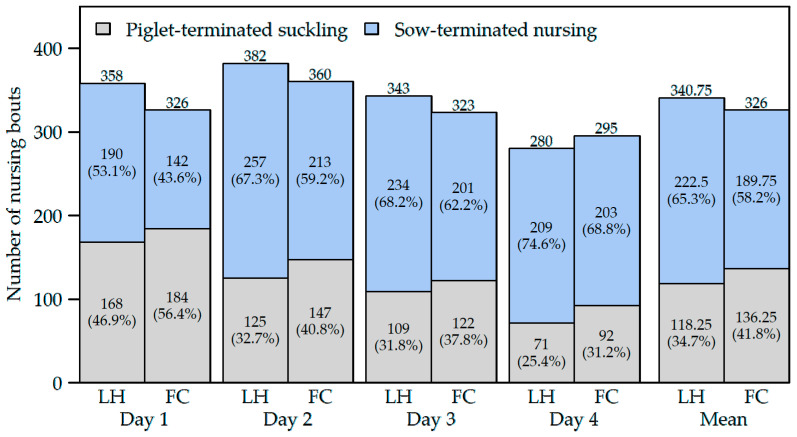
Number and percentages of nursing bouts in the two farrowing systems (loose-housing pens = LH, pens with farrowing crate = FC, each *n* = 30 sows) on the four observation days (one day per week).

**Figure 4 animals-12-00137-f004:**
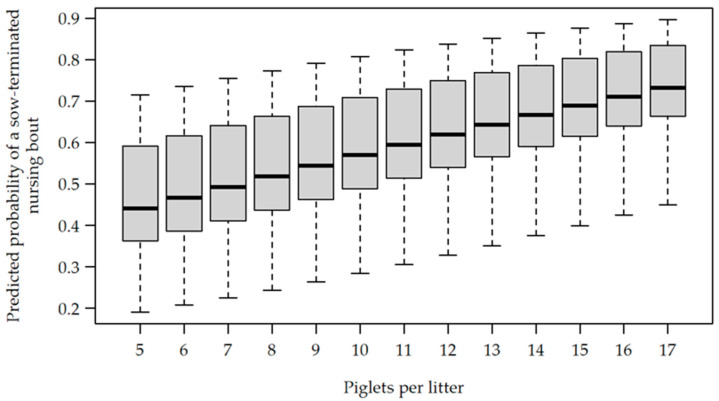
Predicted probability of a sow-terminated nursing bout, depending on the litter size.

**Figure 5 animals-12-00137-f005:**
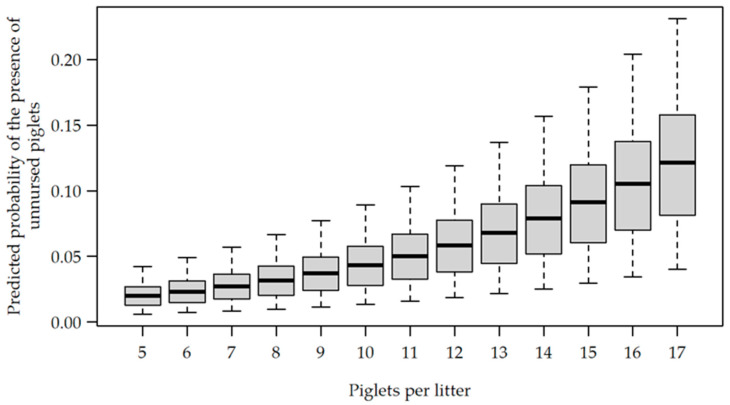
Predicted probability for the presence of unnursed piglets, depending on the litter size.

**Figure 6 animals-12-00137-f006:**
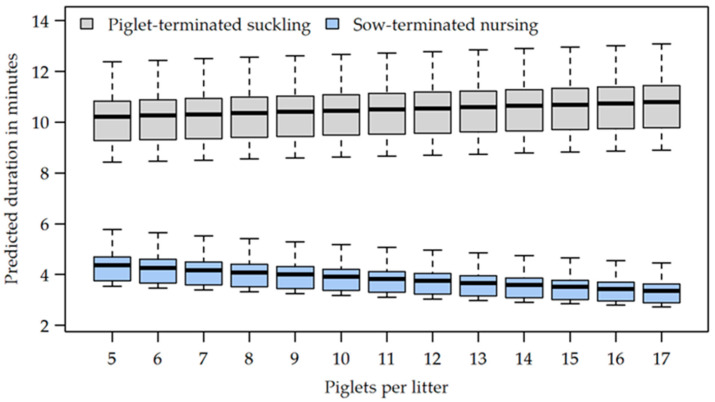
Duration of nursing bouts predicted from the statistical model, depending on the litter size.

**Table 1 animals-12-00137-t001:** Nursing frequency per sow per hour in the two farrowing systems ^1^ on the four observation days ^2^.

System	Day	Nursing Bouts per Sow per Hour
LH	1	1.33 ± 0.84
	2	1.42 ± 0.89
	3	1.22 ± 0.78
	4	1.03 ± 0.69
	Mean	1.25 ± 0.82
FC	1	1.21 ± 0.75
	2	1.33 ± 0.82
	3	1.15 ± 0.70
	4	1.09 ± 0.71
	Mean	1.19 ± 0.75

^1^ Loose housing pens = LH, pens with farrowing crate = FC, each *n* = 30. ^2^ One day per week.

**Table 2 animals-12-00137-t002:** Results of frequencies and durations of nursing bouts in both systems, as well as the percentage of nursing bouts with unnursed piglets. Adjusted *p*-values of pairwise comparisons of the four observation days from the models are marked by * *p* < 0.05 or ** *p* < 0.001.

Day/Week	Nursing Bouts *n*	Mean Duration (Min.)	Sow-Termination *n*	Sow-Terminated (Min.)	Piglet-Terminated (Min.)	Percentage of Nursing Bouts with Unnursed Piglets
1	684 **^2,3,4^	7.20	332 **^2,3,4^ (48.54%)	4.35 **^2,3,4^	9.88 *^4^	10.86 **^4^
2	742 **^1,3,4^	5.99	470 **^1,4^ (63.34%)	3.41 **^1^	10.44	8.47 *^4^
3	666 **^1,2,4^	6.21	435 *^4,^ **^1^ (65,32%	3.74 **^1^	10.85	6.93
4	575 **^1,2,3^	5.90	412 *^3,^ **^1,2^ (71.65%)	3.46 **^1^	12.08 *^1^	3.37 *^2,^**^1^
	2667	6.33	1649 (61.83%)	3.70	10.60	7.58
	(total)	(mean)	(total)	(mean)	(mean)	(mean)

^1,2,3,4^ Superscript numbers indicate on which observation days the respective day was significantly different.

**Table 3 animals-12-00137-t003:** Percentage of nursing bouts with unnursed piglets in the two farrowing systems ^1^ on the four observation days ^2^.

System	Day	Percentage of Nursing Bouts with Unnursed Piglets
LH	1	11.15
	2	10.60
	3	7.23
	4	3.77
	Mean	8.46
FC	1	10.53
	2	5.90
	3	6.56
	4	2.93
	Mean	6.55

^1^ Loose housing pens = LH, pens with farrowing crate = FC, each *n* = 30 sows. ^2^ One day per week.

**Table 4 animals-12-00137-t004:** Results of nursing bouts duration (in minutes) in the two farrowing systems ^1^ on the four observation days during the four-week period ^2^.

System	Day/Week	Sow-Terminated	Piglet-Terminated	Mean
		(Min.)	(Min.)	(Min.)
LH	1	3.93 ± 2.88	8.88 ± 3.91	6.25 ± 4.20
	2	3.27 ± 2.32	9.91 ± 4.68	5.44 ± 4.52
	3	3.59 ± 2.73	10.23 ± 4.88	5.70 ± 4.71
	4	3.23 ± 2.28	11.31 ± 5.24	5.28 ± 4.81
	Mean	3.49 ± 2.56	9.83 ± 4.62	5.69 ± 4.56
FC	1	4.91 ± 2.57	10.79 ± 4.28	8.23 ± 4.66
	2	3.58 ± 2.00	10.89 ± 4.17	6.57 ± 4.73
	3	3.92 ± 3.09	11.40 ± 3.97	6.75 ± 5.00
	4	3.69 ± 2.45	12.67 ± 4.57	6.49 ± 5.29
	Mean	3.95 ± 2.59	11.27 ± 4.27	7.01 ± 4.96

^1^ Loose-housing pens = LH, pens with farrowing crate = FC, each *n* = 30 sows. ^2^ One day per week.

## Data Availability

The data presented in this study are available on reasonable request from the corresponding author. The data are not publicly available due to the large data set.

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
