# Peer review of "Does Nursing Behaviour of Sows in Loose-Housing Pens Differ from That of Sows in Farrowing Pens with Crates?"

_animals, 2022, doi:10.3390/ani12020137_

Round 1
Reviewer 1 Report
Review (animals – 1501927 – Title: Does Nursing Behaviour of Sows in Loose-Housing Pens Differ From That of Sows in Farrowing Pens With Crates?)
This study is important and timely as pork producers are considering of moving from farrowing crates to loose farrowing. In general, the manuscript is well written with many detailed information, except the result section. My major concern is the presentation of the results, and my second concern is the consistency of terminology.
Presentation of Results:
- Tables are not organized in alinement with the text of the results. For instance, Table 1 was explained/described under several sub-sections (3.1; 3.2; 3.3) in the Results which is really confusing and hard for readers to follow and understand. It would be easier for readers to follow if the results in each table are described within one paragraph or sub-section. This is the issue for the entire Results section. Within aach table, it is easier for readers to understand if the table is described in order (such as starting from the first column or first row, instead of starting from the third column - Line 249 for Table 1).
- It appears that each table only lists the descriptive data (means and SD, and %) without statistics, except Table 2. (but the way that p-values are presented in Table 2 is not seen in most scientific journals). Generally, each table should stand alone for readers to understand the data without checking the content of the results. Without statistics, it is very hard for readers to understand the tables (such as which number should be compared with which).
I would suggest to re-design the tables according to the recommendations above. Additionally, please separate the table title from its foot notes. For example, the content in parenthesis of the title of Table 1 should be foot notes. Furthermore, please give a precise title for each table, and correct errors in the tables. For instance, for Table 1, the title should be something like ‘Nursing frequency, number (percentage) of nursing bouts terminated by sows or piglets in two farrowing systems”. Also, in table 1, “Total” does not mean “total” for all variables, such as for % of nursing and nursing frequency.
Terminology: Inconsistent use of terminology can confuse and mislead readers. For instance:
- Litter size, instead of “the number of piglets present in a litter’, should be used throughout the paper.
- Nursing frequency (one defined), instead of “the number of nursing bouts per sow per hour”, should be used throughout the paper
- Nursing duration and bouts instead of “suckling duration” (Line 519) or “suckling bouts” (Line 529)
- Percentage of nursing bouts with unnursed piglets instead of “% With unnursed piglets” (Table 1), “Nursing bouts with unnursed piglets in %” Table 2, or “unnursed litter mates” (Line 311). Regardless which term you choose, please keep it consistent throughout the paper.
- Nursing bouts terminated by the sow, or by piglets, instead of “without terminated by the sow”
- Observation day, instead of ‘the time the sows spent in the farrowing system’ should be used in most cases. This is because ‘observation day’ was included in the statistical models of data analysis and readers are looking for the effect of ‘observation day’ in the results. Additionally, ‘time that sows spent in the farrowing system’ can be misleading for adaptation period to the farrowing system.
Other questions:
- What were the litter sizes (live and total) at birth and weaning for both treatment groups? Were there any differences in litter size at birth or piglet preweaning mortality between the two treatment groups?
- Was there any interactions between observation day and treatment for variables analyzed (nursing frequency, duration, % of nursing bouts with unnursed piglets, % of nursing bouts terminated by the sow or piglets)?
- For all odds ratios reported, please add confidence interval (CI). Without CI, it is hard to convince readers that the difference between the means makes sense. Additionally, when you compare a bigger mean to a smaller mean, readers expect to see OR greater than 1, and vice versa. However, this is opposite in several places of the paper (Lines 28, 29, 273, 324). Finally, when P < 0.05, please add OR and CI (Lines 258, 260, 348).
Reviewer 2 Report
This is a well written article that adds information to a valuable topic area. The introduction in general is well researched and presented, the methods are clear and the results are thorough and, in the main, well presented. The discussion brings in relevant points and concludes well with regards to the findings.
A couple of points below would be good to address:
Line 42: This would be better backed up with several articles to support and indeed there should be reference to the new legislative proposals to ban the farrowing crate system in the EU.
Line 81: should be cited, I assume the citations in the next sentence refer to this study but it should be made clearer.
Results should report the R value of the models that have been run.
As the main finding is the comparison of FH vs LH sow terminated bouts this would have been nice to see visualised in a graph.
Sow terminated bouts over the observation day is reported as increasing but numbers reported in table 1 suggest day 2 is much higher than day 1, can this discrepancy be explained?
If these results were broken down by system this might be easier to follow.
Line 511 as an extra space
How do your results relate to Loftus et al 2020?
Loftus, L., Bell, G., Padmore, E., Atkinson, S., Henworth, A., & Hoyle, M. (2020). The effect of two different farrowing systems on sow behaviour, and piglet behaviour, mortality and growth. Applied Animal Behaviour Science, 232, 105102.
Reviewer 3 Report
BRIEF SUMMARY
The aim of the study was to investigate the nursing behaviour of sows in two different farrowing systems: sows in farrowing pens with crates and sows in a loose-housing system without crates. For that purpose, nursing behaviour of sows was studied with regard to nursing frequency, duration, and termination.
The topic of this paper is highly current, due to the higher requirements for animal welfare on intensive systems of animal production. Its hypothesis that nursing behaviour in loose-housing pens is not impaired compared to farrowing pens with crates is well devised.
BROAD COMMENTS
The experimental trial was performed with a sufficient number of animals, 60 sows, and is well defined, although some considerations could be made:
- There is no indication regarding how the period used for analysis of four days per sow was defined. The number of days does not seem to be very high for a nursing period of 28 days.
- Out of those 4 days, only 8 hours a day were analysed. Only one day of the week (Saturday) is chosen, alleging that there is little disturbance. This may lead to mistaken conclusions, as the aim is to observe behaviour in normal operating conditions, not on days in which conditions are different due to decreased activity. Consequently, results may have been compromised.
- Moreover, as lactation is not limited to the day period, the hours that were chosen for observation (exclusively during the day) should be justified.
The statistical analysis is complete, although the choice of methods should be justified. There should also be an indication regarding whether the methods complied with the cases of application.
In view of the statistical analysis that was performed, conclusions should be much more concrete. Given the conditions in which the sow is on both of the studied housing systems, it does seem logical to assume that sows in LH pens are more likely to exhibit natural nursing behaviour than sows in FC pens. However, conclusions are not justified based on the obtained results and on statistical analysis. Consequently, the conclusions that are drawn were to be expected and are not well defined. It would be advisable to rewrite them.
Round 2
Reviewer 3 Report
Dear Authors,
All my suggestions have been considered and I agree with the changes made. However, I believe that the conclusions must go without references.
Best regards,